# Allergic Reactions to Vaccines in Children: From Constituents to Specific Vaccines

**DOI:** 10.3390/biomedicines11020620

**Published:** 2023-02-18

**Authors:** Ming-Han Tsai, Chih-Yung Chiu

**Affiliations:** 1Department of Pediatrics, Chang Gung Memorial Hospital, Keelung Branch, Keelung 204, Taiwan; 2College of Medicine, Chang Gung University, Taoyuan 333, Taiwan; 3Molecular Infectious Disease Research Center, Chang Gung Memorial Hospital, Taoyuan 333, Taiwan; 4Division of Pediatric Pulmonology, Department of Pediatrics, Chang Gung Memorial Hospital, Taoyuan 333, Taiwan

**Keywords:** allergic reaction, vaccines, children

## Abstract

Vaccination is an essential public health measure that helps to reduce the burden of infectious diseases in children. Although vaccines have an excellent safety record and the association of severe allergic reactions is rare, public concerns about vaccine safety can lead to incomplete vaccination coverage in children with or without allergies. Therefore, it is important to understand the mechanisms and implications of allergic reactions to vaccines and define strategies to manage them to provide the safest care for vaccine recipients. In this review, we provide an overview on the types of allergic reactions that can occur after vaccination, including those caused by various vaccine constituents. We also discuss the mechanisms underlying these allergic reactions and the recommended diagnosis and management strategies for children with a history of suspected allergic reactions to vaccines. An improved understanding of allergic reactions to vaccines can aid in the enhancement of the safety and effectiveness of vaccination.

## 1. Introduction

Vaccination is a highly effective public health intervention that has significantly reduced the morbidity and mortality associated with different infectious diseases [1]. The Advisory Committee on Immunization Practices (ACIP) in the United States recommends a routine immunization schedule in which children receive 10 vaccines for protection against 16 diseases before the age of 2 years [2]. While vaccines can provide protection, vaccine-associated allergic reactions can occur, ranging between one in 50,000 and one in 1,000,000 doses [3]. The most serious reaction is anaphylaxis, which is rare but can occur at a rate of one per 100,000 to one per 1,000,000 doses in the case of commonly administered vaccines [4]. Additionally, when estimating the rate of serious allergic reactions, it is important to consider the systematic use of causality assessment [5]. Causality assessment is the process of determining the relationship between an adverse event and a vaccine. It may help to identify the likelihood that the adverse event was actually caused by the vaccine, rather than by chance or other factors. Despite the low incidence of severe vaccine-associated allergic reactions, the increasing prevalence of other allergic diseases has led to increasing concerns about the possibility of allergic reactions following vaccination [6].

Allergic reactions to vaccines are characterized by an immune-mediated response to one or more components of the vaccine. Moreover, these must be distinguished from other clinical reactions that occur coincidentally with vaccination such as anxiety, vasovagal responses, or local injection-site reactions [7]. These allergic reactions can be triggered by vaccine antigens, residual media used in vaccine production, preservatives, stabilizers, or other excipients [8]. The symptoms of allergic reactions can range from mild cutaneous symptoms (i.e., redness and itching) to more severe multisystem effects (anaphylaxis) that affect the skin, gastrointestinal, respiratory, and cardiovascular systems [9]. Although severe allergic reactions to vaccines may pose a potential risk, the benefits of vaccination outweigh the potential risks to the majority of the population. However, fear of allergic reactions can contribute to vaccine hesitancy, which can compromise herd immunity and hinder efforts to control the spread of infectious diseases [10].

The World Allergy Organization (WAO) proposed a classification system for immunologic reactions to vaccines based on the timing of symptom onset [11]. According to this system, there are two general types of reactions: immediate and delayed reactions. Immediate reactions occur within minutes to one hour after vaccination and are most likely mediated by IgE antibodies. Delayed reactions may appear several hours to days after vaccination. These reactions are rarely mediated by IgE antibodies. This review focuses on immediate-type allergic reactions to vaccines that have the potential to cause life-threatening anaphylaxis in susceptible individuals. We also discuss the various allergic and non-allergic reactions that may occur after vaccination, the components of vaccines that may be responsible for these reactions, and allergic reactions to specific vaccines. The recommended approach for managing patients with a history of suspected allergic reactions to vaccines has also been described.

## 2. Non-Allergic and Allergic Reactions to Vaccination

### 2.1. Non-Allergic Reactions to Vaccines 

Vaccines may be associated with the development of any untoward medical occurrence that arises in conjunction with the use of a drug in humans, regardless of the event being related to the drug [12]. Often referred to as adverse events, these can manifest as local or systemic reactions and can be immune- or non-immune-mediated. Local non-allergic reactions, such as swelling and erythema at the injection site, are common and may occur hours or days after vaccination [10,13]. Systemic nonallergic reactions, including fever and vasovagal reactions, can also occur after vaccination. Vasovagal reactions are characterized by symptoms such as hypotension, pale skin, sweating, weakness, bradycardia, vomiting, and loss of consciousness [14,15]. It can be challenging to distinguish between vasovagal reactions and anaphylaxis because both may present with symptoms such as hypotension and collapse. However, there are some key differences between the two manifestations. Vasovagal reactions are often preceded by pale skin, whereas anaphylaxis may start with flushing and may also include symptoms such as itching, hives, and swelling [16]. Anaphylaxis is more likely to cause tachycardia, whereas a vasovagal reaction may cause bradycardia. It is important to accurately differentiate between a systemic vasovagal reaction and anaphylaxis during vaccination to ensure that appropriate treatment is promptly administered.

### 2.2. Allergic Reactions to Vaccines

Allergic reactions to vaccines can range in severity and may involve only a few symptoms or manifestations across multiple organs. Typical symptoms and signs of an allergic reaction to vaccines include rhinoconjunctivitis, bronchoconstriction, gastrointestinal symptoms, and skin lesions such as generalized hives and/or angioedema [15]. Symptoms of an allergic reaction to a vaccine may appear alone or in combination and typically occur within minutes to four hours after vaccination. Children with mild allergic reactions to vaccines may experience swelling and itching at the injection site, conjunctivitis, or a runny nose [7]. Those with moderate allergic reactions may experience bronchoconstriction or generalized hives. Severe allergic reactions to vaccines are rare but can be life-threatening and may require hospitalization [9]. These reactions can cause significant incapacitation, congenital anomalies, or even death.

Anaphylaxis is a severe life-threatening allergic reaction that can affect multiple organ systems [10]. The incidence of anaphylaxis after vaccination is rare, with rates ranging from 0.3 to 2.1 per million vaccine doses based on active surveillance studies [17,18]. Anaphylaxis can affect the skin and respiratory, gastrointestinal, and cardiovascular systems, and symptoms may include generalized hives, wheezing, swelling of the mouth, tongue, and throat, difficulty in breathing, vomiting, diarrhea, low blood pressure, decreased levels of consciousness, and shock [19,20]. The National Institute of Allergy and Infectious Disease/Food Allergy and Anaphylaxis Network symposium has proposed the diagnostic criteria for anaphylaxis [21]. These include one of the following three scenarios: (1) the sudden onset of symptoms within minutes or hours involving the skin and/or mucous membranes (itching, flushing, hives, and angioedema) and respiratory compromise (difficulty in breathing, wheezing/bronchospasm, stridor, and low oxygen levels) or decreased blood pressure/end-organ dysfunction (collapse, syncope, and incontinence); (2) two or more of the following symptoms occurring rapidly after exposure to a likely allergen for the patient: skin and/or mucosa symptoms; respiratory compromise; decreased blood pressure/end organ dysfunction; persistent gastrointestinal symptoms (vomiting, abdominal pain, and diarrhea); and (3) reduced blood pressure within minutes or hours after exposure to a known allergen.

Immediate allergic reactions to vaccines are caused by the presence of allergen-specific immunoglobulin E (IgE) that can bind to high-affinity FcεRI receptors on mast cells and basophils [22]. When an allergen is subsequently encountered, the cell-bound IgE antibodies can be cross-linked, triggering the degranulation of mast cells and/or basophils, thereby triggering the release of histamine and other chemical mediators (cytokines, interleukins, leukotrienes, and prostaglandins) into the surrounding tissue. This can cause a range of systemic effects, including vasodilation, mucous secretion, tissue eosinophilic infiltration, and airway smooth muscle contraction [10,23]. The differences between vasovagal reactions and anaphylaxis are shown in Table 1.

## 3. Allergic Reactions to Vaccine Constituents

Unlike drugs, which are the primary cause of immediate allergic reactions, vaccine excipients (i.e., substances used to formulate a vaccine) are the main cause of specific IgE and immediate reactions associated with vaccines. Pre-existing allergies to vaccine excipients, such as antigens, adjuvants, stabilizers, preservatives, emulsifiers, leached packaging components, residual antibiotics, cell culture materials, and inactivating ingredients, are the primary contributors to immediate allergic reactions during vaccination [22]. In contrast, drug-related anaphylaxis is usually caused by the active drug itself rather than excipients.

### 3.1. Gelatin

Gelatin is used as a stabilizer in several vaccines. It has been identified as a cause of many anaphylactic reactions to measles, mumps, rubella (MMR), and varicella vaccines [24,25]. A retrospective study that collected sera from individuals who had experienced anaphylaxis after receiving the MMR vaccine found that 27% of them had anti-gelatin IgE, whereas none of the vaccinated subjects who did not experience adverse events had detectable levels of this antibody [26]. Further research has shown that patients who experience anaphylaxis due to the MMR vaccine may also be sensitized to gelatin present in the diphtheria–tetanus–acellular pertussis (DTaP) vaccine [27]. Gelatin is also a source of alpha-gal, a carbohydrate allergen that causes meat allergies. Children with an alpha-gal allergy may experience anaphylaxis after receiving an MMR or varicella vaccine [28].

Thus, children with allergic reactions to gelatin upon ingestion should be evaluated by an allergist before receiving gelatin-containing vaccines. If there is a history of an immediate-type allergic reaction to gelatin, which is confirmed by skin tests or serum-specific IgE antibody tests, it is advisable to test such children for gelatin-containing vaccines prior to administration. If the vaccine skin tests are negative, the vaccine can be administered in the usual manner. However, children should be observed for at least 30 min after vaccine administration. If the vaccine skin tests are positive, the vaccine can be administered in incremental doses under observation [7].

### 3.2. Egg

Egg proteins are present in some influenza, MMR, and yellow fever vaccines because they are cultured in chicken embryo fibroblasts or embryonated chicken eggs [10,29]. Egg allergy is common in children, but studies have shown that influenza and MMR vaccines can be safely administered to children with egg allergies [30,31]. In particular, children with egg allergies, including those who experienced anaphylaxis, successfully received yellow fever vaccines without any serious adverse events [32]. According to the ACIP guidelines, individuals with mild egg allergies can receive any licensed age-appropriate influenza vaccine and no longer need to be observed for 30 min after vaccination, as severe allergic reactions to these vaccines are rare [10]. However, individuals with severe egg allergies should only receive influenza vaccines under the supervision of a healthcare provider capable of recognizing and managing serious allergic conditions. For children with egg allergies, the current practice for the yellow fever vaccine is to undergo skin testing before administration. If the vaccine skin test results are negative, the vaccine can be administered in the usual manner. However, the children should be observed for at least 30 min. If the skin test results are positive, the vaccine can be administered safely in graded doses [33].

### 3.3. Milk Protein 

Casein, a protein found in cow milk, has been implicated in anaphylaxis to DTP-containing vaccines in a small number of children with severe milk allergies [34]. These vaccines are prepared using a medium derived from cow milk protein, and trace amounts of casein are found in these preparations. However, there is no evidence to suggest that DTP vaccines contribute to the development of allergic diseases or that atopy is a contraindication for these vaccines [19].

### 3.4. Preservatives/Adjuvants (Thimerosal, Aluminum, and Phenoxyethanol)

Thimerosal, aluminum, and phenoxyethanol are added to some vaccines as preservatives. However, thimerosal, which contains mercury, is rarely used as a preservative in vaccines, and its role as an allergen remains unclear [35]. A risk assessment study found no evidence of harm caused by thimerosal in vaccines, except for local hypersensitivity reactions [36]. Additionally, it should be noted that thimerosal is a mercury-based organic compound, and therefore does not possess any toxic characteristics for mammals. These preservatives are not known to cause immediate-type allergic reactions and immediate-type skin testing is not necessary.

### 3.5. Antibiotics

Antibiotics, including neomycin, polymyxin B, kanamycin, gentamicin, and streptomycin, can cause allergic reactions ranging from mild to severe, including anaphylaxis. For example, an individual who received the MMR vaccine containing neomycin was reported to have experienced anaphylaxis shortly after vaccination [37]. In addition, one case of anaphylaxis has been reported following the use of eye drops containing polymyxin B, an excipient found in DTaP, and other vaccines [38]. To the best of our knowledge, no other antibiotic has been linked to vaccine-associated anaphylaxis. A commonly used preservative in cosmetics and vaccines is 2-phenoxyethanol. There are favorable reports about this compound for its broad-spectrum antimicrobial activity and good tolerability [10,39].

### 3.6. Latex

The use of rubber in vaccine vial stoppers or syringe plungers can cause allergic reactions in some individuals. A report of an anaphylactic reaction following the administration of the hepatitis B vaccine to a latex-allergic patient has been attributed to the rubber in the vaccine stopper [40]. However, a review of 160,000 reports from the Vaccine Adverse Event Reporting System (VAERS) found only 28 cases of possible immediate-type allergic reactions following the administration of a vaccine containing dry natural rubber [41]. Patients with a history of anaphylaxis to latex can safely receive vaccines with non-latex packaging. However, if the only available option contains latex, the vaccine can still be administered. In such cases, the patient must be observed for at least 30 min afterward.

### 3.7. Yeast 

There are relatively few reported instances of anaphylaxis following vaccination in individuals with known allergies to yeast proteins. Between 1990 and 2004, only 15 such cases were reported, 11 of which occurred after administration of the hepatitis B vaccine. This vaccine contained trace amounts of yeast proteins [10]. According to a report from VAERS, 107 adverse events were reported in individuals with a pre-existing yeast allergy. Of them, 11 were probable or possible anaphylactic events following administration of the hepatitis B vaccine [42]. It has been reported that individuals with yeast allergies may experience reactions to vaccines containing yeast proteins. If a patient has a history of reaction to baker’s or brewer’s yeast and has a positive skin test for *Saccharomyces cerevisiae*, it is recommended to test with yeast-containing vaccines before administration [7,15]. If the vaccine skin test result is negative, the vaccine can be administered as usual, with the patient being observed for at least 30 min. If the vaccine skin test is positive, the vaccine can be administered in incremental doses, with the patient being monitored.

### 3.8. Dextran 

The use of dextran as a medium nutrient or stabilizer in some vaccines, including the MMR vaccine previously used in Italy and Brazil, has been linked to allergic reactions [43]. These reactions were attributed to the presence of IgG antibodies to dextran and activation of the complement system, leading to the release of anaphylatoxins. This brand of MMR vaccine has been removed from the market. However, dextran can still be found in some other vaccines such as rotavirus vaccines.

### 3.9. Polyethylene Glycol 

The Pfizer-BioNTech BNT162B2 and the Moderna mRNA-1273 vaccines are lipid nanoparticles that contain messenger ribonucleic acid (mRNA) coding for the spike protein of the coronavirus [10]. Lipid nanoparticles stabilize and improve the solubility of mRNA vaccines in water and act as adjuvants. These vaccines contain polyethylene glycol (PEG) 2000, a high-molecular-weight version of PEG. This is used as an emulsifier in a variety of products, including vaccines, pharmaceuticals, cosmetics, and foods. PEG and its derivatives are commonly found in household products [10]. There is evidence that sensitivity to PEG may lead to IgE-mediated anaphylaxis after the administration of PEG-conjugated biologics, and severe allergic reactions to PEG have been linked to pre-existing anti-PEG antibodies induced by PEG-containing household products. Although more research is needed to understand the higher rate of anaphylaxis observed with COVID-19 vaccines compared to other vaccines, PEG 2000, a component of mRNA vaccines such as Pfizer-BioNTech BNT162B2 and Moderna mRNA-1273, is considered the most likely cause of allergic reactions. Here, Table 2 shows the components implicated in allergic reactions and related adverse events.

## 4. Allergic Reactions to Specific Vaccines

### 4.1. Diphtheria, Tetanus, and Pertussis Vaccines

Immediate allergic reactions to routine vaccines are rare, with an estimated incidence of 2 per million doses of DTaP [17]. If the primary vaccination is with an acellular pertussis vaccine, specific IgE antibodies to D, T, and P vaccines may be common after booster doses. In addition, elevated levels of pertussis toxin IgE have been associated with local reactions [47,48]. Casein, a protein found in cow milk, has been identified as a potential cause of anaphylaxis in children with severe milk allergies or high levels of specific milk IgE following vaccination with DTP-containing vaccines [34]. However, in the case of DTP-containing vaccines, there have been reports of anaphylaxis caused by the presence of casein. It is important to note that these cases are rare and that most children with severe milk allergies can tolerate these vaccines. Thus, no change is prescribed in vaccine recommendations [8,19].

### 4.2. Influenza Vaccine 

Influenza vaccines, such as trivalent and quadrivalent inactivated influenza vaccines (IIVs), recombinant subunit influenza vaccines (RIV), and live attenuated influenza vaccines (LAIVs), are considered safe for individuals with asthma. Studies have shown that IIVs do not pose a significant risk to individuals with asthma, including those with severe disease [49]. A marginal increase in the risk of medically significant wheezing was observed in children aged 6–23 months who received LAIVs. However, no increased risk was found in children aged 2–5 years [50]. Additionally, a Cochrane review found no significant increase in acute asthma in adults or children older than 3 years following IIVs [51]. Furthermore, the available data support the safety and efficacy of LAIVs in children aged 2–17 years with mild to moderate asthma [19,52].

Previous studies have provided evidence that IIVs, which contain low levels of ovalbumin (<0.12 μg/mL), can be administered safely to patients with egg allergy, including those with severe reactions [53]. Des Roches et al. conducted a systematic review of 26 studies involving a total of 4172 patients, among whom 513 had a history of severe allergic reactions to eggs. The authors found no evidence of anaphylaxis associated with influenza vaccination [54]. Similarly, a more recent review of 28 studies comprising a total of 4315 subjects with egg allergy, including 656 with a prior history of anaphylaxis following egg ingestion, reported no serious adverse reactions after influenza vaccination [55]. However, data on the safety of LAIVs in egg allergy have been a cause of concern. Studies such as SNIFFLE have found no systemic reactions. There are reports of a small percentage (1.6%) of mild, self-limiting reactions in individuals with egg allergy receiving LAIV, including those with a history of anaphylaxis to eggs [53,56]. As a result of these findings, the current UK immunization recommendations no longer consider egg allergy a contraindication to LAIV, unless the individual has experienced life-threatening anaphylaxis that requires intensive care treatment [19].

### 4.3. Measles, Mumps, and Rubella Vaccine 

It has been determined that anaphylactic reactions to MMR vaccines are typically a result of gelatin allergies [57]. Prior to the administration of MMR vaccines to individuals with egg allergies, studies have been conducted to assess the safety of vaccines in these individuals. The results of these studies have shown that MMR vaccination is safe for children with egg allergies [58]. This was exemplified in a study of 54 unvaccinated children with confirmed egg allergies, where none of the participants experienced any adverse reactions to the vaccine [59]. Additionally, in the Danish childhood vaccination programme, no reactions were observed in 32 patients who were sensitized to hen’s eggs after receiving the MMR vaccine [60]. The British Society for Allergy and Clinical Immunology (BSACI) guidelines for the management of egg allergy recommend that children with egg allergies receive routine MMR vaccination in primary care [19,61].

### 4.4. Hepatitis B Vaccine 

The hepatitis B vaccine is manufactured using yeast cells. Thus, residual *Saccharomyces cerevisiae* antigens might be present in the product. While there have been reports of anaphylaxis in association with the hepatitis B vaccine [58], these cases have not been confirmed through skin tests or measurement of allergen-specific IgE levels in serum. A review of the literature on adverse reactions to the hepatitis B vaccine indicated that the incidence of anaphylaxis is less than 1 in 100,000 vaccinations [62]. 

### 4.5. Pneumococcal and Meningococcal Vaccines

According to current evidence, there is generally no contraindication to administering the pneumococcal vaccine to children with allergies. This excluded a single reported case of anaphylaxis that was attributed to the non-toxic diphtheria CRM197 conjugating protein present in the 13-valent pneumococcal conjugate vaccine [63]. With an estimated incidence of one per million doses administered, the incidence of anaphylactic reactions to meningococcal polysaccharide or polysaccharide–protein conjugate vaccines is indeed considered to be extremely rare [64]. One study describes the findings of post-marketing surveillance of adverse events following meningococcal B vaccination in Italy from 2014 to 2019 [65]. A total of 214 cases of adverse events were reported, resulting in an incidence rate of 26.5 per 100,000 doses. The cases were categorized as serious (27.1%), not serious (67.8%), and undefined (5.1%). Among the 58 serious cases, 31 (53.4%) were deemed to have a ‘consistent causal association’ with the meningococcal B vaccine. Of these cases, the most common adverse event was fever/hyperpyrexia, which was reported in 21 of the 31 cases (67.7%), while a hypotonic–hyporesponsive episode was reported in 7 of the 31 cases (22.6%).

### 4.6. COVID-19 Vaccine 

The Centers for Disease Control and Prevention (CDC) recommend COVID-19 vaccines, including Pfizer-BioNTech BNT162B2 and Moderna mRNA-1273, for everyone 6 months and older. Both vaccines have undergone rigorous testing and have been shown to be safe and effective in preventing COVID-19 [66]. The reported rates of anaphylaxis after COVID-19 vaccination vary depending on the surveillance method used. For example, passive reporting has resulted in rates of 2.5 to 4.7 events per million doses [67], active surveillance has resulted in rates of 4.8 to 5.1 events per million doses [68], and a meta-analysis report reported a rate of 7.91 events per million doses [66]. Available evidence suggests that the incidence of anaphylaxis because of COVID-19 vaccination is higher among females [69,70]. According to some studies, most anaphylactic reactions occurred within 30 min of receiving the Pfizer-BioNTech vaccine and within 15 min of receiving the Moderna vaccine [65,66,67]. No fatalities have been reported [70].

Some individuals may have allergic reactions to certain components of COVID-19 vaccines, such as polyethylene glycol (PEG). In cases where children have an allergic reaction to these components or to the first dose of the vaccine, it is important to discuss the available options with a healthcare provider and allergist. Subsequently, it is important to make a shared decision on vaccination. Currently, it is recommended that individuals who have had an allergic reaction to the first dose of the COVID-19 vaccine should not receive the second dose. Further research is needed to clarify the safety and accuracy of skin testing for allergies to COVID-19 or PEG [70]. Moreover, there is a dearth of data on the tolerance of individuals who have had allergic reactions to COVID-19 vaccines to future vaccinations with different COVID-19 vaccines [71]. To date, there is no conclusive evidence that polyethylene glycol (PEG), an excipient in both mRNA COVID-19 vaccines (Pfizer-BioNTech and Moderna), is responsible for allergic reactions to these vaccines. More research is needed to understand the relationship between PEG and allergic reactions to COVID-19 vaccines and to determine the safety and efficacy of vaccination for individuals who have previously experienced allergic reactions.

## 5. Approach to Children with a Suspected Vaccine Allergy

When evaluating children with suspected allergic reactions to vaccines, it is important to consider several key points to make informed decisions about future vaccinations. One important aspect to consider is whether the nature and timing of the reaction are consistent with an IgE-mediated reaction, such as hives, angioedema, and difficulty in breathing. These typically occur within minutes after vaccination. Another point to consider is obtaining a history of similar reactions to the same or other vaccines or to vaccine constituents. This may help to identify any specific components or ingredients that are responsible for the reaction [7]. The third point to consider when evaluating children with suspected allergic reactions to vaccines is determining the need for future doses of a particular vaccine or other vaccines that contain similar components. Even if the child does not require additional doses of the vaccine in question, an allergic reaction could indicate a potential hypersensitivity to a component that is also present in other vaccines. The child might be required to receive such vaccines in the future. An overall approach to treating children with suspected vaccine allergic reactions is shown in Figure 1.

### 5.1. Skin Testing 

If a child with a suspected allergic reaction to a vaccine is scheduled to receive additional doses of the same or a related vaccine, it is important to perform skin testing before administering the vaccine [19]. Skin testing is used to determine whether a person is sensitized to a specific allergen, in this case, a component of the vaccine. However, it is important to note that proper performance and interpretation of skin tests require expertise in the procedure, including the use of appropriate positive and negative controls. All tests should be performed by allergists with training in the interpretation and treatment of possible reactions to the tests. Additionally, skin testing should be performed in a setting where anaphylactic reactions can be quickly recognized and treated and where proper resuscitation equipment is available [10].

Skin testing can provide additional information about sensitization to certain allergens and help determine the likelihood that a specific allergen is responsible for an allergic reaction. The testing procedure typically begins with a skin-prick test using an undiluted vaccine. A positive reaction indicates an allergic response. If the full-strength skin prick test is negative, an intradermal test using a 1:100 dilution of the vaccine in isotonic (0.9%) saline is typically performed, and appropriate positive and negative controls are used to ensure accurate test results [15]. It is important to note that false-positive reactions can occur during skin testing, particularly at a dilution of 1:10, in cases of influenza, MMR, and varicella vaccines. False-positive reactions have been reported at a dilution of 1:100 in a small percentage of controls for DTaP (5%) and influenza (15%) [72]. This highlights that positive skin test results should be considered indicative rather than confirmatory, and additional testing or evaluation may be required.

Skin testing can also be performed for specific components of a vaccine that are potentially allergenic, such as eggs, gelatin, latex, and yeast. Commercially available skin test reagents are available for eggs and yeast. For gelatin testing, an extract can be prepared by dissolving one teaspoon of sugared gelatin powder in 5 mL of normal saline and using it as a prick skin test solution [7]. In addition to skin testing, levels of specific IgE can also be measured in serum using commercially available immunoassays.

### 5.2. Management of Allergic Reactions to Vaccines 

Children who have been sensitized to a vaccine or its components and have experienced an anaphylactic reaction to the vaccine should only be revaccinated if it is necessary. While previous allergic reactions to vaccines are a concern, it is important to note that most cases with prior allergic reactions to vaccines can be safely vaccinated in the future with appropriate precautions. These include processes such as skin testing, premedication, and close observation during vaccination [73].

The management of allergic reactions to vaccines is primarily based on the results of skin testing. If a child’s intradermal test with the vaccine is negative, it is highly unlikely that they have an IgE antibody to any of the vaccine constituents. In such cases, the vaccine can be administered in the usual manner [7]. It is also important to observe children who have had a previous allergic reaction to vaccines for at least 30 min after administration. Additionally, it is important to ensure the ready availability of epinephrine and other treatments in case of an allergic reaction.

If skin tests for a vaccine or its components are positive, but the vaccine is deemed necessary, the vaccine may still be administered using a graded-dose protocol [7]. This protocol typically involves administering a small initial dose (10%) of the vaccine and waiting for 30 min to assess the development of any allergic reactions. If no allergic reaction occurs, the remaining 90% of the dose is subsequently administered [7,19]. As an alternative approach to the graded-dose protocol, rapid desensitization has been used to reduce the risk of anaphylaxis while administering a vaccine to an individual with a history of allergic reactions to its components. This method involves administering incremental doses of the vaccine at 15 min intervals. However, this is permissible only when there are no signs of an allergic reaction. The schedule and the dose of the vaccine used in this method may vary, but it generally starts with a very low dose of a dilution of the vaccine, such as 0.05 mL of a 1:10 dilution. The dose is gradually increased to 0.05 mL, 0.1 mL, 0.15 mL, 0.20 mL, and 0.5 mL [15]. Rapid desensitization has been successfully used for many vaccine-allergic individuals but has not been specifically evaluated for COVID-19 vaccines [10]. While the use of a rapid desensitization method or graded-dose protocol can help reduce the risk of an anaphylactic reaction, it is important to note that there is still a risk of an allergic reaction. Thus, sufficient caution must be exercised while using such methods. This method should only be undertaken after obtaining informed consent from the patient and/or their parents. Additionally, the decision to use this method should be made in consultation with an allergist or another healthcare professional who is experienced in the management of allergic reactions and desensitization protocols [7].

## 6. Conclusions

Vaccination is an important public health measure that helps in the reduction of the spread of infectious diseases and protects the health of individuals and communities. However, it is possible for allergic reactions to occur because of vaccination. Anaphylactic reactions to vaccines typically present with symptoms that are similar to those of anaphylaxis from other causes, which usually appear within minutes to an hour following vaccination. However, in rare cases, the onset of symptoms may be delayed up to several hours. Moreover, it is important to note that most of the allergic reactions to vaccines are IgE-mediated. This indicates that they are caused by the immune system’s overreaction to certain components of the vaccine, rather than the actual microorganisms against which the vaccine is designed. 

Individuals who are at risk of developing allergic reactions to vaccines or who have a history of allergic responses to vaccines must be evaluated by an allergist. The evaluation process for a possible vaccine allergy begins by determining if the symptoms and timing of the reaction are consistent with an anaphylactic reaction. The second step is to discern whether the child had previous exposure to the vaccine or if they need subsequent doses of the vaccine or vaccines with common constituents in the future. Clinical testing, including skin-prick and intradermal tests, can help identify allergens in at-risk individuals. These tests can include testing individual vaccine components to determine the antigen that causes an allergic reaction. If skin testing for a component or the vaccine shows a positive reaction, it may still be possible for the child to receive the vaccine using a graded-dose protocol. However, it is important to reiterate that this should only be undertaken after a detailed assessment of the relative risks and benefits of vaccination by the child’s healthcare provider, an allergist, and the child’s parents.

Increased understanding of allergic reactions to vaccines can help improve the manufacturing process and safety of vaccines. By identifying the specific components that cause allergic reactions, vaccine manufacturers can remove them or replace them with alternate compounds. This would help in decreasing the overall risk of allergic reactions to vaccines. Such an understanding also has an important impact on the management of vaccination as it allows the development of targeted approaches for individuals who are at a higher risk of an allergic reaction. Furthermore, we consider that effective scientific communication between clinicians and the general population is important. Addressing people’s concerns and fears should be one of the priorities for clinicians, as doing so can increase confidence not only in vaccination but also in medical science in general. 

## Figures and Tables

**Figure 1 biomedicines-11-00620-f001:**
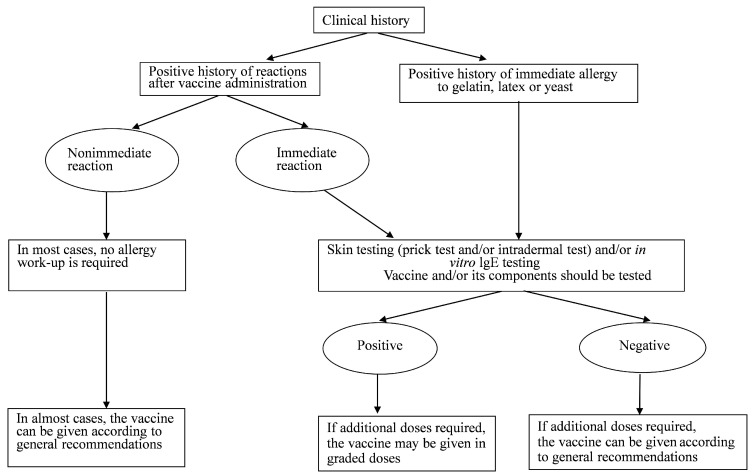
Management of children with suspected hypersensitivity to a vaccine, and children with a known allergy to a vaccine component (modified from Dreskin, et al. 2016; reprinted/adapted with permission from Elsevier) [7].

**Table 1 biomedicines-11-00620-t001:** Differentiation between vasovagal reaction and anaphylactic reaction.

Characteristics	Vasovagal Reaction	Anaphylactic Reaction	Ref.
Onset	Immediately or shortly after injection	Within minutes, typically within 30 minutes	[10,15]
Respiratory	Normal, possible hyperventilation, but not labored	Cough, wheeze, stridor, rhinorrhea, respiratory distress signs (e.g., tachypnea, cyanosis, and subcostal retraction)	[19]
Cardiovascular	Self-limited bradycardia, absent peripheral pulse, but with a strong central pulse Hypotension, usually transient and corrected in the supine position Loss of consciousness, improves once in the supine position	Tachycardia and absent central pulse Hypotension, sustained and no improvement without adequate management Loss of consciousness and no improvement once in the supine position	[19,20]
Skin	Pale, sweaty, and cool clammy skin	Flushing, itchy rash, angioedema, and urticaria	[10,19]
Gastrointestinal	Nausea or vomiting	Abdominal cramps	[10]
Neurological	Self-limited loss of consciousness and good response to prone positioning	Altered consciousness and poor response to prone positioning	[19]

**Table 2 biomedicines-11-00620-t002:** Immediate component-mediated reactions and associated vaccines in children.

Component	Function in Vaccines	Relevant Vaccines	Allergic Reactions	Ref.
Gelatin	Stabilizer	MMR and varicella	Anaphylaxis and urticaria	[24,25]
Egg protein (albumin)	Residual medium and stabilizer	MMR, influenza, and yellow fever	Anaphylaxis	[10,29]
Milk protein (casein)	Medium nutrient	DTaP	Anaphylaxis	[34]
Thimerosal	Preservative	Influenza and Td	Local reaction	[35,36]
Aluminum	Adjuvant	DTaP, Hib, hepatitis A/B, HPV, Japanese encephalitis, meningococcal, pneumococcal, and Tdap	Local reaction	[10]
Neomycin	Antimicrobial	DTaP, hepatitis A, influenza, MMR, polio, and varicella	Anaphylaxis	[38]
Phenoxyethanol	Preservative	DTaP, influenza, polio, and Tdap	Local reaction	[39]
Latex	Pharmaceutical closure	DTaP, hepatitis A/B, influenza, meningococcal, rotavirus, Tdap, Td, and yellow fever	Anaphylaxis and urticaria	[40,41]
Yeast	Medium nutrient	DTaP, hepatitis B, HPV, meningococcal, and pneumococcal	Anaphylaxis	[10,42]
Dextran	Medium nutrient and stabilizer	MMR ^a^ and rotavirus	Anaphylaxis	[43]
Polyethylene glycol	Surfactant of mRNA	COVID-19	Anaphylaxis	[44,45,46]

^a^ MMR vaccines containing dextran have been withdrawn from the market. MMR, measles–mumps–rubella; DTaP, diphtheria–tetanus–acellular pertussis; Td, tetanus–diphtheria; Tdap, tetanus-diphtheria-acellular pertussis; Hib, *Hemophilus influenzae* type b; HPV, human papillomavirus.

## Data Availability

Not applicable.

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
