# Peer review of "Allergic Reactions to Vaccines in Children: From Constituents to Specific Vaccines"

_biomedicines, 2023, doi:10.3390/biomedicines11020620_

Round 1

Reviewer 1 Report

This is an excellent review of a topic sometimes neglected, but very  important particularly in the current times. the authors have prepared a nice table listing potential allergens that contaminate vaccines and a flow chart for addressing children's potential vaccine allergies. There is one slight typo in one of the boxes of that chart - just a missing word.

Otherwise good to go!

Author Response

Point 1: This is an excellent review of a topic sometimes neglected, but very important particularly in the current times. the authors have prepared a nice table listing potential allergens that contaminate vaccines and a flow chart for addressing children's potential vaccine allergies. There is one slight typo in one of the boxes of that chart - just a missing word.

Otherwise good to go!

Response 1: Thank you for the reviewer’s comments. After English spelling examination, the words of the boxes in Figure 1 have been carefully checked and amended.

Reviewer 2 Report

”. After reading it, I have the following comments.

1.     The review is exhaustive and well written, addresses an interesting topic and is easy to read and understand. I only have minor remarks, most of which are required by the excellent level of the paper.

2.     Introduction: Authors should consider the importance of systematic use of causality assessment to estimate the real incidence of serious allergic adverse event

·       doi:   10.1155/2022/2911333.

·       doi: 10.1016/j.vaccine.2018.01.018

3.     Page 2, lines 42-48: a reference is needed.

4.     Page 4, line 48: thimerosal has been the main protagonist of a long debate regarding vaccine safety, due to toxicity concerns raised by pseudo-scientific no-vax groups. In consideration of that, the authors should specify that thimerosal is an organic compound of mercury, and as such holds no toxic properties for mammal life forms.

5.     Page 5, line 33-39: a reference is needed.

6.     Page 7, paragraph 2: data of safety profile of influenza vaccine should be implemented

·       doi: 10.7774/cevr.2015.4.2.137

7.     Page 8, line 1-4: data of safety profile of meningococcal vaccine should be added

·       doi: 10.1080/21645515.2021.1963171

8.     Page 8, line 5: “Centers for Disease Control and Prevention” is plural, and therefore requires “recommend” instead of “recommends”.

9.     The paper’s Conclusions section should mention the fundamental role of scientific information communication in counteracting vaccination hesitancy. Addressing the subject’s fears should be one of the priorities of vaccination specialists, as by overcoming them it is possible to increase trust towards not only vaccination practices, but also towards medical science as a whole.

Author Response

Point 1: The review is exhaustive and well written, addresses an interesting topic and is easy to read and understand. I only have minor remarks, most of which are required by the excellent level of the paper

Response 1: Thank you for the reviewer’s comments. The responses to your comments were shown as the following.

Point 2: Introduction: Authors should consider the importance of systematic use of causality assessment to estimate the real incidence of serious allergic adverse event

  • doi: 10.1155/2022/2911333.
  • doi: 10.1016/j.vaccine.2018.01.018

Response 2: Thank you for the reviewer’s comments. It is important to consider the systematic use of causality assessment when estimating the incidence of serious allergic adverse events. Causality assessment is the process of determining the relationship between an adverse event and a drug or other invervention. It may help to identify the likelihood that the adverse event was actually caused by the drug, rather than by chance or other factors.

   A systematic approach to causality assessment can improve the accuracy of incidence estimates and provide more reliable data for decision-making in the healthcare field. It also helps to identify potential risk factors and to inform the development of preventative measures.

   We added the related description in the revised manuscript (page 1, line 35-39).

Point 3: Page 2, lines 42-48: a reference is needed.

Response 3: Thank you for the reviewer’s comments. In the revised manuscript, references 7 and 9 were added in page 2, line 49 and page 3, line 1, respectively.

Point 4: Page 4, line 48: thimerosal has been the main protagonist of a long debate regarding vaccine safety, due to toxicity concerns raised by pseudo-scientific no-vax groups. In consideration of that, the authors should specify that thimerosal is an organic compound of mercury, and as such holds no toxic properties for mammal life forms.

Response 4: Thank you for the reviewer’s comments. As the reviewer’s suggestion, the sentences “Additionally, it should be noted that thimerosal is a mercury-based organic compound, and therefore does not possess any toxic characteristics for mammals” were added in page 5, line 6-7.

Point 5: Page 5, line 33-39: a reference is needed.

Response 5: Thank you for the reviewer’s comments. References 7 and 15 were added in page 5, line 42.

Point 6: Page 7, paragraph 2: data of safety profile of influenza vaccine should be implemented

  • doi: 10.7774/cevr.2015.4.2.137

Response 6: Thank you for the reviewer’s comments. The data of safety profile of influenza vaccine and related references (Ref 55, 56) were added in page 7, line 23-28.

Point 7: Page 8, line 1-4: data of safety profile of meningococcal vaccine should be added

  • doi: 10.1080/21645515.2021.1963171

Response 7: Thank you for the reviewer’s comments. The data of safety profile of meningococcal vaccine and related reference (Ref 66) was added in page 8, line 12-20.

Point 8: Page 8, line 5: “Centers for Disease Control and Prevention” is plural, and therefore requires“recommend” instead of “recommends”.

Response 8: Thank you for the reviewer’s comments. The word “recommends” was revised as “recommend” (page 8, line 23).

Point 9: The paper’s Conclusions section should mention the fundamental role of scientific information communication in counteracting vaccination hesitancy. Addressing the subject’s fears should be one of the priorities of vaccination specialists, as by overcoming them it is possible to increase trust towards not only vaccination practices, but also towards medical science as a whole.

Response 9: Thank you for the reviewer’s comments. The sentences “Furthermore, we considered that the effective scientific communication between clinicians and general population is important. Addressing people’s concerns and fears should be one of the priorities for clinicians, as doing so can increase confidence not only in vaccination, but also in medical science in general” were added in page 11, line 7-10.
